

# Bacterial diversity and potential risk factors associated with *Salmonella* contamination of seafood products sold in retail markets in Bangkok, Thailand

Edward R. Atwill[1] and Saharuetai Jeamsripong[2]

[1] School of Veterinary Medicine, University of California, Davis, Department of Population Health and Reproduction, Davis, CA, USA
[2] Research Unit in Microbial Food Safety and Antimicrobial Resistance, Department of Veterinary Public Health, Faculty of Veterinary Science, Chulalongkorn University, Bangkok, Thailand

## ABSTRACT

Consumption of contaminated food causes 600 million cases, including 420,000 of fatal infections every year. Estimated cost from food-borne illnesses is USD 110 billion per year, which is an economic burden to low- and middle-income countries. Thailand is a leading producer and consumer of seafood, but little is known about bacterial contamination in seafood. In particular, public health agencies need to know the relationship between *Salmonella* contamination in seafood and risk factors, as assessed with readily available culture-dependent and bacterial phenotyping methods. To address this, levels of indicator bacteria, *Salmonella* and *Vibrio* in various seafood products were determined to identify risk factors associated with *Salmonella* contamination. A total of 335 samples were collected from October 2018 to July 2019 at seafood markets throughout Bangkok, Thailand; overall sample composition was Pacific white shrimp ($n = 85$), oysters ($n = 82$), blood cockles ($n = 84$), and Asian seabass ($n = 84$). Prevalence was 100% for fecal coliforms and 85% for *E. coli*. In contrast, prevalence was 59% for *V. parahaemolyticus*, 49% for *V. cholerae*, 19% for *V. alginolyticus*, 18% for *V. vulnificus*, and 36% for *Salmonella*. Highest concentrations of fecal coliforms and *E. coli* were in oysters. Highest concentrations of *Salmonella* with Matopeni (31%) being the predominant serotype were in shrimp. *Salmonella* contamination was significantly associated with type of seafood, sampling location, retail conditions, and the presence of *E. coli*, *V. alginolyticus* and *V. vulnificus*. A cutoff value for *E. coli* concentration of $1.3 \times 10^4$ MPN/g predicted contamination of *Salmonella*, with a sensitivity of 84% and specificity of 61%. Displaying seafood products on ice, presence of *E. coli* and *Vibrio*, and seafood derived from Eastern Thailand were associated with an increased risk of *Salmonella* contamination.

Corresponding author
Saharuetai Jeamsripong,
saharuetai.j@chula.ac.th

## INTRODUCTION

The global fisheries and aquaculture both inland and marine reached 171 million tonnes in 2016 (*Food and Agriculture Organization of the United Nations, 2018*) and consumption of fish, and fishery products *per capita* double from 10 kg in 1960 to greater than 20 kg in 2016 (*Food and Agriculture Organization of the United Nations, 2018*). In Southeast Asia, consumption of fish and fishery products varies from 6 to 64 kg *per capita* per year (*Food and Agriculture Organization of the United Nations, 2015*). In Thailand, consumption of fish and fishery products is about 31 kg *per capita* per year, which accounts for 12% of total protein consumption per person (*Food and Agriculture Organization of the United Nations, 2015*). Thailand is one of the top ten exporters of fish and fishery products, which accounted for 4% of global exports in 2016 (*Food and Agriculture Organization of the United Nations, 2018*).

Due to the rapid growth of global consumption of fish and fishery products, seafood safety is of high public health concern. Foodborne diseases afflict a third of the world population each year (*World Health Organization, 2004*), but data about the number of illnesses from seafood-borne outbreaks is limited for many parts of the world. Most examinations of seafood outbreaks have been done in the United States, where approximately 9.4 million illnesses, almost 56,000 hospitalizations, and 1,351 deaths, were associated with foodborne contamination per year (*Scallan et al., 2011*). Almost half (45%) of foodborne outbreaks reported in the U.S. are from bacteria, and fish are frequently implicated (*Gould et al., 2013*). In Europe, 5,175 foodborne outbreaks were reported in 2019. *Salmonella* spp. cause most of these outbreaks. There were 87,923 and 7,775 confirmed cases of salmonellosis and infections from Shiga-toxin-producing *Escherichia coli*, respectively (*European Food Safety Authority; European Centre for Disease Prevention & Control, 2021*).

Pollution, animal density, and global trading contribute to bacterial contamination of seafood products (*Papadopoulou et al., 2007*). The most common pathogens associated with seafood-borne diseases are *Vibrio*, *Salmonella*, *Shigella*, and *Clostridium botulinum* (*Iwamoto et al., 2010*). Seafood-borne outbreaks caused by *V. parahaemolyticus*, *V. cholera* serogroup O139, *V. vulnificus*, *Salmonella* serotype Weltevreden, and *E. coli* have been reported (*Bonnin-Jusserand et al., 2019*; *Heinitz et al., 2000*; *Martinez-Urtaza et al., 2016*; *Raymond & Ramachandran, 2019*).

In Thailand, *V. parahaemolyticus* and *Salmonella* spp. are the leading causes of foodborne diarrhea. Even though Thailand is one of the major exporters of seafood products, monitoring of bacterial pathogens of these exports is limited. Therefore, the objectives of this study were: (1) examine the prevalence of bacterial indicators, *V. parahaemolyticus*, *V. cholerae*, *V. vulnificus*, *V. alginolyticus*, and *Salmonella* isolated from Pacific white shrimp, oysters, blood cockle, and Asian seabass in fresh markets in Bangkok, Thailand; (2) identify serotypes of *Salmonella* among various seafood samples; and (3) determine risk factors for *Salmonella* contamination and a cutoff value for *E. coli* concentration that predicts the presence of *Salmonella* in fresh seafood.

**Table 1 Factors associated with bacterial contamination of seafood products sold in Bangkok, Thailand.**

| Risk factors | No. of sample (%)[1] | | | | |
| --- | --- | --- | --- | --- | --- |
| | Pacific white shrimp | Oyster | Blood cockle | Asian seabass | Total |
| **Type of seafood** | 85 (25.4%) | 82 (24.5%) | 84 (25.1%) | 84 (25.1%) | 335 (100%) |
| **Sampling district** | | | | | |
| Din Daeng | 11 (30.6%) | 7 (19.4%) | 4 (11.1%) | 14 (38.9%) | 36 (100%) |
| Huay Kwang | 17 (22.4%) | 21 (27.6%) | 24 (31.6%) | 14 (18.4%) | 76 (100%) |
| Samphanthawong | 28 (25.5%) | 26 (23.6%) | 28 (25.5%) | 28 (25.5%) | 110 (100%) |
| Dusit | 29 (25.7%) | 28 (24.8%) | 28 (24.8%) | 28 (24.8%) | 113 (100%) |
| **Regional source of seafood** | | | | | |
| Central | 65 (25.3%) | 53 (20.6%) | 72 (28.0%) | 60 (26.1%) | 257 (100%) |
| Eastern | 3 (10.7%) | 0 (0%) | 16 (57.1%) | 9 (32.1%) | 28 (100%) |
| Southern | 10 (40.0%) | 6 (24.0%) | 5 (20.0%) | 4 (16.0%) | 25 (100%) |
| Unidentified | 7 (28.0%) | 7 (28.0%) | 7 (28.0%) | 4 (16.0%) | 25 (100%) |
| **Retail storage** | | | | | |
| Pool | 4 (4.9%) | 0 (0%) | 0 (0%) | 78 (95.1%) | 82 (100%) |
| Separate | 81 (32.0%) | 82 (32.4%) | 84 (33.2%) | 6 (2.4%) | 253 (100%) |
| **Retail display condition** | | | | | |
| Without ice | 39 (31.2%) | 28 (22.4%) | 47 (37.6%) | 11 (8.8%) | 125 (100%) |
| On ice | 46 (21.9%) | 54 (25.7%) | 37 (17.6%) | 73 (34.8%) | 210 (100%) |

**Note:**
[1] Row totals of percentages may not add to 100% during to rounding of decimals.

## MATERIALS AND METHODS

### Sample collection

Samples of fresh fish and shellfish ($n = 335$) were collected from open-air retail fresh markets between October 2018 and July 2019 from four districts in Bangkok, Thailand, resulting in a sample composition of Pacific white shrimp (*Litopenaeus vannamei*) ($n = 85$), oyster (*Saccostrea cuccullata*) ($n = 82$), blood cockle (*Tegillarca granosa*) ($n = 84$), and Asian seabass (*Lates calcarifer*) ($n = 84$) (Table 1). Due to varying availability of these four different seafood commodities at each market, there were slightly different total sample sizes for some seafood commodities ranging from $n = 82$ to $n = 85$ (Table 1). Pacific white shrimp, oysters, and blood cockles are raised in saltwater ponds; the majority of Asian seabass are raised in estuaries, but some are raised in saltwater ponds.

Individual seafood samples were purchased in the early morning (5 to 7 a.m.). At least 200 g of the samples were placed into a double sterile plastic bag. The samples were kept on ice (<10 °C) during transportation and kept in the cooler. All samples were submitted to the laboratory within 3 h. Microbiological determination was performed within 6 h after receiving samples in the Department of Veterinary Public Health, Faculty of Veterinary Science, Chulalongkorn University.

Average and standard deviation (sd) of minimum and maximum ambient air temperature (°C), wind speed (km/h), precipitation (mm), and relative humidity (%) in Bangkok, Thailand, were retrieved from the Thai Meteorological Department

(www.tmd.go.th). The average (± sd) daily minimum and maximum ambient air temperature was 26.8 (± 1.8) °C and 34.1 (± 2.6) °C; average (± sd) wind speed was 13.0 (± 2.0) km/h, average 24-hour precipitation 1.4 (± 4.0) mm, and average relative humidity was 75.1 (± 7.4)%.

## Predictor variables

Risk factors for *Salmonella* contamination included type of seafood (Pacific white shrimp, oyster, blood cockle, or Asian seabass), sampling district (Din Daeng, Huay Kwang, Samphanthawong, or Dusit), regional source of seafood (central, eastern, southern Thailand, or unidentified source), retail storage of fish and shellfish samples (pooling and combining different seafood products for retail display *versus* keeping each seafood type separate when on display), and retail display condition (on ice or without ice). The concentrations of fecal coliform (MPN/g) and *E. coli* (MPN/g), and the prevalence of *V. parahaemolyticus*, *V. vulnificus*, *V. alginolyticus* and *V. cholerae* were evaluated as putative risk factors for *Salmonella* contamination.

## Bacterial concentration and phenotyping

Seafood samples were analyzed in triplicate for coliforms, *E. coli*, *Salmonella*, *V. parahaemolyticus*, *V. vulnificus*, *V. alginolyticus* and *V. cholerae*. Fecal coliform and *E. coli* were enumerated according to the U.S. Food and Drug Administration (U.S. FDA) Bacteriological Analytical Manual (BAM) with slight modification (*Feng et al., 2002*). Briefly, a 25 g sample (shrimp and Asian seabass) was aseptically cut into small pieces and placed into 225 mL of Buffered Peptone Water (BPW) (Difco, Sparks, MD, USA). Pieces were then homogenized for 1 to 2 min. The resulting suspension was serially diluted using three-tube most probable number (MPN) at different dilutions from $10^{-1}$ to $10^{-4}$. One mL of each solution was diluted in Lactose Broth (LB) (Difco, Sparks, MD, USA), and then incubated at 35 °C for 24 h. A loopful of the mixture solution was transferred to Brilliant Green Lactose Bile (BGLB) (Difco, Sparks, MD, USA) and EC broth (Difco, Sparks, MD, USA), respectively. After overnight incubation, positive tubes were recorded and calculated as concentration of fecal coliforms (MPN/g). One loopful of EC broth was streaked on Eosin Methylene Blue (EMB; Difco, Sparks, MD, USA) agar plates and reported as *E. coli* concentration (MPN/g).

Fecal coliform and *E. coli* were enumerated in oysters and blood cockles following (*Feng et al., 2002*). Briefly, 100 g of meat was added into 100 mL of Phosphate Buffered Saline (PBS) (Difco, Sparks, MD, USA), which was then blended aseptically for 1 to 2 min. The resulting suspension was serially diluted in LB to $10^{-4}$. These dilutions were transferred to BGLB and EC broth. Biochemical tests including indole test and Triple Sugar Iron (TSI; Difco, Sparks, MD, USA) were performed on suspect colonies for all samples. The lower and upper limits of the detection of fecal coliforms and *E. coli* were 1.0 and $1.1 \times 10^5$ MPN/g, respectively.

*Salmonella* detection followed ISO 6579-1:2017 (*International Organization for Standardization, 2017*). Briefly, 25 g of seafood was cut, and added to 225 mL of BPW. The pieces were homogenized for 2 min and incubated at 37 °C for 18 h. After incubation,

0.1 mL of the suspension was inoculated into Modified Semi-solid Rappaport-Vassiliadis (MSRV) (Difco, Sparks, MD, USA) agar plate and incubated at 42 °C overnight. A loopful from the MSRV plates was restreaked onto Xylose Lysine Deoxycholate (XLD) (Difco, Sparks, MD, USA) agar. Presumptive colonies of *Salmonella* were pink to red colonies with a black center. Biochemical tests (citrate utilization, TSI reaction, indole test) were used to confirm presumptive *Salmonella* colonies according to a standard protocol from the U.S. FDA BAM (*Andrews et al., 2007*). Three typical colonies of *Salmonella* were selected for serotyping. Slide agglutination test was performed to determine serotype of *Salmonella* followed by Kauffmann–White Scheme, Pasteur Institute (*Grimont & Weill, 2007*). Commercial antiserums (S&A Reagents Lab Ltd., Lat Phrao, Bangkok, Thailand) were used to determine the serotype of *Salmonella*.

Isolation of *Vibrio* spp. followed U.S. FDA BAM (*Kaysner, DePaola & Jones, 2004*). Briefly, 50 g of each sample was added to 450 mL of PBS, and homogenized for 1 to 2 min. One mL of resulting suspension was added to 10 mL of Alkaline Peptone Water (APW) (Difco, Sparks, MD, USA) and incubated at 37 °C overnight. After incubation, one loopful of solution was streaked on Thiosulfate-Citrate-Bile Salts-sucrose (TCBS) (Difco, Sparks, MD, USA) agar plate containing 2% of NaCl. Presumptive colonies of *Vibrio* were confirmed using CHROMagar™ *Vibrio* (HiMedia Laboratories, Mumbai, India) agar. TCBS and CHROMagar™ *Vibrio* plates were incubated at 37 °C for 24 h. Colonies with green center on TCBS agar were presumed to be *V. parahaemolyticus*. Colorless colonies were presumed to be *V. vulnificus*. On CHROMagar™ *Vibrio* agar plate, mauve colonies were presumed to be *V. parahaemolyticus*, and green blue to turquoise blue were presumed to be of *V. vulnificus*. Colorless colonies were presumed to be *V. alginolyticus*.

Isolation of *V. cholerae* followed U.S. FDA BAM (*Kaysner, DePaola & Jones, 2004*). Briefly, 25 g of sample was added to 225 mL of APW, homogenized for 1 to 2 min, and incubated at 35 ± 2 °C for 8 h. A loopful of solution was streaked to TCBS agar plates. After incubation at 37 °C for 24 h, presumptive colonies of *V. cholerae* were confirmed on CHROMagar™ *Vibrio*. Typical colonies of *V. cholerae* on TCBS agar plate are 2 to 3 mm diameter, yellow, and flat colonies with opaque center, whereas the presumptive colonies of *V. cholerae* in CHROMagar™ *Vibrio* agar were green blue to turquoise blue. Biochemical tests including TSI, oxidase test, and growth in sodium chloride were conducted to confirm *Vibrio* identifications.

## Statistical analyses

Chi-square test and odds ratios were used to examine the association between different species of bacterial contamination and different types of seafood. For the odds ratio calculations of the association between bacteria contamination among seafood samples, shrimp was set as the referent category based on its popularity in Thai cuisine and largest sample size ($n = 85$) of the four seafood commodities. In addition, logistic regression was used to determine the association between *Salmonella* contamination and various risk factors. To construct the final logistic regression model, univariate associations were first evaluated for all risk factors for *Salmonella* and an initial multivariable model constructed

**Table 2 Concentrations of fecal coliforms and *E. coli* in seafood products sold in Bangkok, Thailand.**

| Samples | Fecal coliforms | | E. coli | |
| --- | --- | --- | --- | --- |
| | Prevalence (%) | Average ± sd (MPN/g) | Prevalence (%) | Average ± sd (MPN/g) |
| Shrimp ($n = 85$) | 85 (100%) | $9.43 \times 10^4$ ($3.4 \times 10^4$) | 85 (100%) | $1.07 \times 10^4$ ($2.5 \times 10^4$) |
| Oyster ($n = 82$) | 82 (100%) | $1.10 \times 10^5$ ($7.1 \times 10^3$) | 82 (100%) | $5.13 \times 10^4$ ($4.5 \times 10^3$) |
| Blood cockle ($n = 84$) | 84 (100%) | $5.71 \times 10^4$ ($4.8 \times 10^4$) | 68 (81.0%) | $5.85 \times 10^3$ ($2.2 \times 10^4$) |
| Asian seabass ($n = 84$) | 84 (100%) | $8.70 \times 10^4$ ($4.1 \times 10^4$) | 56 (66.7%) | $1.30 \times 10^3$ ($2.6 \times 10^3$) |
| Total ($n = 335$) | 335 (100%) | $8.70 \times 10^4$ ($4.1 \times 10^4$) | 285 (85.1%) | $1.85 \times 10^4$ ($3.7 \times 10^4$) |

**Note:**
sd, standard deviation.

from only significant univariate risk factors ($P \leq 0.2$). A backward stepping algorithm was then used to eliminate non-significant ($P > 0.05$) risk factors based on a likelihood ratio test resulting in a final multivariable logistic regression model with only significant ($P \leq 0.05$) risk factors. Receiver operating characteristic (ROC) analysis was performed to predict contamination of *Salmonella* using estimation of the concentration of *E. coli*. Based on ROC analysis, the optimal cutoff value for the concentration of *E. coli*. was determined. All statistical analyses were performed using Stata version 14.0 (StataCorp, College Station, TX, USA). A $P$-value < 0.05 was considered as statistically difference under the two-sided hypothesis test.

## RESULTS

### Occurrence of indicator bacteria in seafood samples

All seafood products sampled in Bangkok were positive for fecal coliforms with total average concentration (± sd) at $9 \times 10^4$ (± $4 \times 10^4$) MPN/g (Table 2). The prevalence of *E. coli* was 85%, with total average concentration (± sd) of $2 \times 10^4$ (± $4 \times 10^4$) MPN/g. Oyster samples had the highest concentrations (± sd) of fecal coliforms at $1 \times 10^5$ (± $7 \times 10^3$) and *E. coli* at $5 \times 10^4$ (± $5 \times 10^3$), while blood cockle and seabass had the lowest concentrations of these indicator bacteria (Table 2).

### Occurrence of *Vibrio* and *Salmonella* in seafood samples

The overall prevalence for the various bacterial pathogens observed in all 335 seafood samples was 59% for *V. parahaemolyticus*, 18% for *V. vulnificus*, 19% for *V. alginolyticus*, 49% for *V. cholerae* and 36% for *Salmonella*. The highest prevalence of *V. parahaemolyticus*, *V. vulnificus*, *V. alginolyticus*, *V. cholerae*, and *Salmonella* were observed in blood cockle (78%), Pacific white shrimp (33%), oyster (29%), Asian seabass (76%), and Pacific white shrimp (40%), respectively (Fig. 1). The lowest prevalence (<10%) of *V. vulnificus* was observed in blood cockle and for *V. alginolyticus* in Asian seabass. Moreover, shrimp were most likely to have any of the four pathogens, followed by oysters (Fig. 1). Blood cockles exhibited very high contamination of *V. parahaemolyticus*, while Asian seabass tended to harbor *V. cholerae* (Fig. 1). Based on Chi-square tests, the was a significant association between different types of samples and the occurrence of

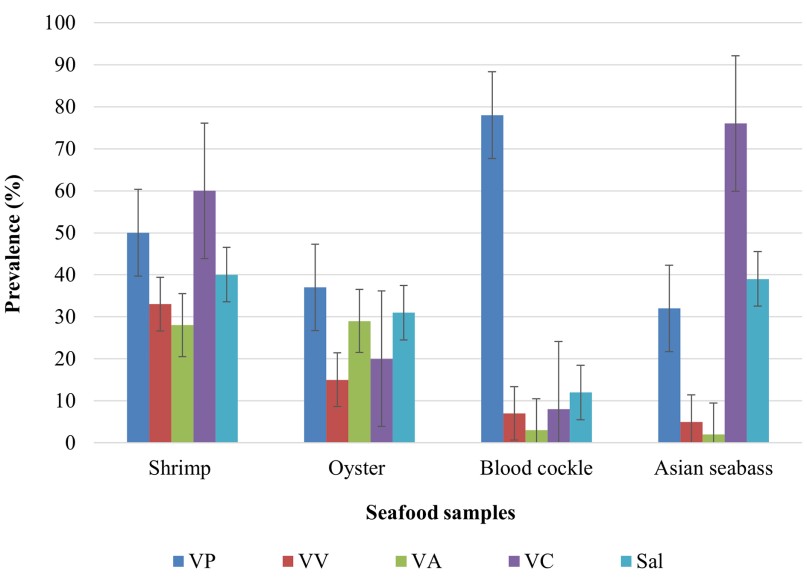

**Figure 1 The prevalence of *V. parahaemolyticus* (VP), *V. vulnificus* (VV), *V. alginolyticus* (VA), *V. cholerae* (VC), and *Salmonella* spp. in shrimp (*n* = 85), oyster (*n* = 82), blood cockle (*n* = 84), and Asian seabass (*n* = 84).**

*V. parahaemolyticus*, *V. vulnificus*, *V. alginolyticus*, *V. cholerae*, and *Salmonella* contamination (*P* < 0.0001).

Pacific white shrimp exhibited a high prevalence of *V. parahaemolyticus* (59%), *V. cholerae* (53%) and *Salmonella* (47%), whereas oysters were mainly contaminated with *V. parahaemolyticus* (45%) and *Salmonella* (38%). In blood cockles, they exhibited a low prevalence of *V. cholerae*, *V. vulnificus*, and *V. alginolyticus*. Asian seabass exhibited a high prevalence of *V. cholerae* (91%) and *Salmonella* (46%).

Matopeni (31%), Corvallis (5%), Give (5%), and Rissen (5%) were the most common serotypes of *Salmonella* isolated from seafood products (Table 3). Matopeni was the predominant serotype (52/56) observed from Asian seabass samples (*n* = 56 isolates), whereas Itami and Leith were common serovars isolated from the shrimp samples (*n* = 47 isolates). For oysters (*n* = 43 isolates) and blood cockles (*n* = 24 isolates), the major serotypes were Give (19%) and Rissen (33%), respectively.

## The distribution of *Salmonella*, *V. parahaemolyticus*, *V. vulnificus*, *V. cholerae*, and *V. alginolyticus* among seafood products

Odds of *V. vulnificus* contamination in shrimp was 7.0 (1/0.143) times higher than that for blood cockle (*P* = 0.002) (Table 4). Odds of *V. cholerae* contamination in shrimp were 7.5 (1/0.134) and 23.6 (1/0.043) times higher than for oyster (*P* < 0.0001) and blood cockle (*P* = 0.002), respectively. The presence of *V. parahaemolyticus* in the blood cockle was higher than in shrimp (OR = 9.1, *P* < 0.0001). The odds of *V. alginolyticus* contamination in shrimp was 13.3 (1/0.075) and 20.0 (1/0.050) times higher than in blood cockles and seabass, respectively.

**Table 3 Distribution of *Salmonella* serovars in seafood products sold in Bangkok, Thailand (*n* = 170 isolates).**

| Serotype | Number of positive (%) | | | | |
|---|---|---|---|---|---|
| | Shrimp | Oyster | Blood cockle | Asian seabass | Total |
| Agona | – | – | 4 (2.35) | – | 4 (2.35) |
| Aminatu | 2 (1.18) | – | – | – | 2 (1.18) |
| Australia | 1 (0.59) | – | – | – | 1 (0.59) |
| Bardo | – | 4 (2.35) | – | – | 4 (2.35) |
| Bonames | 3 (1.76) | – | – | – | 3 (1.76) |
| Breukelen II | – | 4 (2.35) | – | – | 4 (2.35) |
| Corvallis | – | 4 (2.35) | 4 (2.35) | – | 8 (4.71) |
| Dabou | – | 4 (2.35) | – | – | 4 (2.35) |
| Dresden | 1 (0.59) | – | – | – | 1 (0.59) |
| Enteritidis | 3 (1.76) | – | – | – | 3 (1.76) |
| Farmingdale | – | 4 (2.35) | – | – | 4 (2.35) |
| Give | – | 8 (4.71) | – | – | 8 (4.71) |
| Glidji | – | 2 (1.18) | – | – | 2 (1.18) |
| Hisingen | 2 (1.18) | – | – | – | 2 (1.18) |
| Itami | 5 (2.94) | – | – | – | 5 (2.94) |
| Kentucky | 3 (1.76) | – | – | – | 3 (1.76) |
| Konongo | 1 (0.59) | – | – | – | 1 (0.59) |
| Lansing | 1 (0.59) | – | – | – | 1 (0.59) |
| Leith | 4 (2.35) | – | – | – | 4 (2.35) |
| Lexington | – | 4 (2.35) | – | – | 4 (2.35) |
| Lezennes | – | – | 4 (2.35) | – | 4 (2.35) |
| Linguere | 2 (1.18) | – | – | – | 2 (1.18) |
| Matopeni | – | – | – | 52 (30.59) | 52 (30.59) |
| Oslo | 3 (1.76) | – | –– | – | 3 (1.76) |
| Paratyphi B2 | – | – | – | 4 (2.35) | 4 (2.35) |
| Rissen | – | – | 8 (4.71) | – | 8 (4.71) |
| Ruiru | – | 4 (2.35) | – | – | 4 (2.35) |
| Stanley | – | – | 4 (2.35) | – | 4 (2.35) |
| Stuttgart | 2 (1.18) | – | – | – | 2 (1.18) |
| Soerenga | – | 1 (0.59) | – | – | 1 (0.59) |
| Tounouma | 1 (0.59) | – | – | – | 1 (0.59) |
| Typhimurium | 1 (0.59) | – | – | – | 1 (0.59) |
| Victoria | 3 (1.76) | – | – | – | 3 (1.76) |
| Windermere | 3 (1.76) | – | – | – | 3 (1.76) |
| Weltevreden | – | 4 (2.35) | – | – | 4 (2.35) |
| Wohlen | 2 (1.18) | – | – | – | 2 (1.18) |
| Yeerongpilly | 1 (0.59) | – | – | – | 1 (0.59) |
| II/IIIa/IV | 3 (1.76) | – | – | – | 3 (1.76) |
| Total | 47 (27.65) | 43 (25.29) | 24 (14.12) | 56 (32.94) | 170 (100.00) |

**Table 4 Comparing the odds of contamination from *Salmonella*, *V. parahaemolyticus*, *V. vulnificus*, *V. cholerae*, or *V. alginolyticus* among seafood products sold in Bangkok, Thailand.**

| Parameters | OR | S.E. | 95% C.I. | P-value |
|---|---|---|---|---|
| *Salmonella* | | | | |
| Shrimp[1] | 1.0 | – | – | – |
| Oyster | 0.684 | 0.155 | [0.439–1.065] | 0.093 |
| Blood cockle | 0.188 | 0.388 | [0.125–0.281] | <0.0001 |
| Asian seabass | 0.975 | 0.767 | [0.209–4.555] | 0.974 |
| Constant | 0.889 | 0.394 | [0.373–2.121] | 0.791 |
| *V. parahaemolyticus* | | | | |
| Shrimp[1] | 1.0 | – | – | – |
| Oyster | 0.576 | 0.096 | [0.413–0.802] | 0.001 |
| Blood cockle | 9.100 | 1.686 | [6.329–13.085] | <0.0001 |
| Asian seabass | 0.431 | 0.276 | [0.122–1.515] | 0.189 |
| Constant | 1.429 | 0.277 | [0.977–2.088] | 0.065 |
| *V. vulnificus* | | | | |
| Shrimp[1] | 1.0 | – | – | – |
| Oyster | 0.353 | 0.196 | [0.119–1.045] | 0.060 |
| Blood cockle | 0.143 | 0.088 | [0.043–0.477] | 0.002 |
| Asian seabass | 0.100 | 0.130 | [0.008–1.285] | 0.077 |
| Constant | 0.635 | 0.325 | [0.233–1.731] | 0.374 |
| *V. cholerae* | | | | |
| Shrimp[1] | 1.0 | – | – | – |
| Oyster | 0.134 | 0.060 | [0.056–0.322] | <0.0001 |
| Blood cockle | 0.043 | 0.045 | [0.006–0.329] | 0.002 |
| Asian seabass | 3.958 | 5.020 | [0.329–47.548] | 0.278 |
| Constant | 2.400 | 1.596 | [0.652–8.835] | 0.188 |
| *V. alginolyticus* | | | | |
| Shrimp | 1.0 | – | – | – |
| Oyster[1] | 1.114 | 0.418 | [0.534–2.322] | 0.774 |
| Blood cockle | 0.075 | 0.087 | [0.008–0.715] | 0.024 |
| Asian seabass | 0.050 | 0.042 | [0.009–0.260] | <0.0001 |
| Constant | 0.491 | 0.131 | [0.292–0.827] | 0.007 |

**Note:**
[1] Referent category, based on shrimp's popularity in Thai cuisine and largest sample size among the four seafood commodities. OR, Odds ratio. S.E., standard error. C.I., confidence interval.

## Risk factors associated with *Salmonella* contamination

*Salmonella* contamination of seafood sold throughout Bangkok was significantly associated with type of seafood, sampling district, retail display condition, regional source of seafood, and the presence of *E. coli*, *V. alginolyticus*, and *V. vulnificus* (Table 5). *Salmonella* contamination in Pacific white shrimp was not different form Asian seabass; in contrast, both oysters and blood cockles had significantly lower odds of *Salmonella* contamination compared to shrimp. Seafood from markets in Huay Kwang (OR = 1.7) and Dusit (OR = 1.4) had a higher odds of *Salmonella* contamination compared to seafood from Din Daeng and Samphanthawong. Seafood displayed on ice (OR = 1.7, *P* < 0.0001)

**Table 5 Final multivariable logistic regression model for risk factors associated with *Salmonella* contamination of seafood products sold in Bangkok, Thailand.**

| Parameters | OR | S.E. | 95% C.I. | *P*-value |
|---|---|---|---|---|
| **Type of seafood** | | | | |
| Shrimp[1] | 1.0 | – | – | – |
| Oyster | 0.48 | 0.089 | [0.331–0.689] | <0.0001 |
| Blood cockle | 0.19 | 0.045 | [0.119–0.303] | <0.0001 |
| Asian seabass | 1.06 | 0.638 | [0.328–3.450] | 0.919 |
| **Sampling district** | | | | |
| Din Daeng[1] | 1.0 | – | – | – |
| Huay Kwang | 1.66 | 0.232 | [1.259–2.182] | <0.0001 |
| Samphanthawong | 0.43 | 0.046 | [0.351–0.532] | <0.0001 |
| Dusit | 1.42 | 0.120 | [1.205–1.676] | <0.0001 |
| **Retail display condition** | | | | |
| No ice[1] | 1.0 | – | – | – |
| On ice | 1.71 | 0.201 | [1.360–2.154] | <0.0001 |
| **Regional source of seafood** | | | | |
| Central[1] | 1.0 | – | – | – |
| Eastern | 3.46 | 0.670 | [2.325–5.141] | <0.0001 |
| Southern | 0.89 | 0.061 | [0.780–1.020] | 0.094 |
| Unidentified | 0.81 | 0.135 | [0.581–1.120] | 0.199 |
| **Presence of *E. coli*** | | | | |
| No[1] | 1.0 | – | – | – |
| Yes | 4.02 | 1.223 | [2.213–7.295] | <0.0001 |
| **Presence of *V. alginolyticus*** | | | | |
| No[1] | 1.0 | – | – | – |
| Yes | 1.37 | 0.212 | [1.008–1.851] | 0.044 |
| **Presence of *V. vulnificus*** | | | | |
| No[1] | 1.0 | – | – | – |
| Yes | 0.61 | 0.130 | [0.405–0.930] | 0.021 |
| **Constant** | 0.19 | 0.051 | [0.109–0.318] | <0.0001 |

Notes:
AIC = 368.67.
[1] Reference group. OR, Odds ratio. S.E., standard error. C.I., confidence interval. AIC, Akaike information criterion.

had a higher odds of *Salmonella* contamination compared to retail seafood products not displayed on ice. Seafood products sourced from Eastern Thailand had significantly higher odds of *Salmonella* contamination compared to seafood sourced from other regions (OR = 3.5, *P* < 0.0001). Lastly, the odds of *Salmonella* contamination were positively associated with the presence of *E. coli* and *V. alginolyticus*, but negatively associated with *V. vulnificus*.

## ROC and area under the ROC curve

The area under the ROC curve (AUC) at 64% with standard error = 0.30 (CI [58–70%]) (Fig. 2). The ROC AUC was statistically significance (*P* < 0.0001) compared to the null

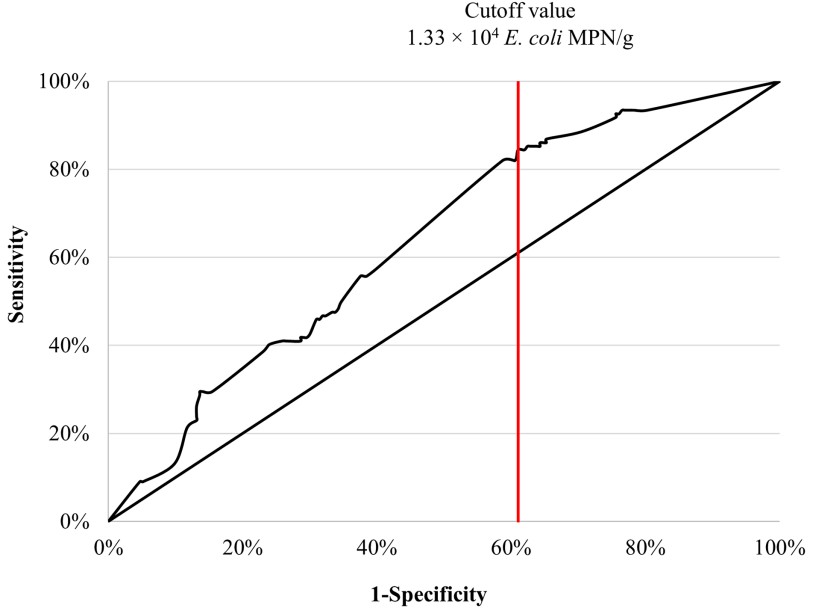

**Figure 2** Area under the ROC curve and cutoff value for concentration of *E. coli* were used to maximally predict the contamination of *Salmonella* in retail seafood commodities (*n* = 335).

value of AUC = 0.5. The presence of *Salmonella* in seafood products was predicted by a concentration of $1.3 \times 10^4$ *E. coli* MPN/g, with a sensitivity of 84% and specificity of 61%.

## DISCUSSION

According to the Ministry of Public Health for Thailand, the concentration of *E. coli* should not exceed 10 MPN/g of fresh or frozen seafood and less than 3 MPN/g of seafood consumed raw; in addition, all products must not contain detectable *Salmonella*, *V. cholerae*, *V. parahaemolyticus* in a 25 g sample (*Bureau of Quality & Safety of Food, 2020*). In this study, the concentration of *E. coli* averaged $2 \times 10^4$ MPN/g for all the seafood samples. In fact, only 18% (*n* = 60/335) of all seafood samples had concentrations of *E. coli* <10 MPN/g and only 7% (*n* = 6/82) of oyster samples (often eaten raw) had <3 MPN/g. Furthermore, the prevalence of *Salmonella* (36%), *V. cholerae* (49%), *V. parahaemolyticus* (59%) indicated widespread bacterial contamination of these seafood products, which also violates food safety standards. Therefore, implementation of basic sanitation and evaluation of microbiological contamination of seafood products sold in Bangkok are needed.

Salmonella is an important pathogen that is responsible for seafood-borne illness worldwide (*Barrett et al., 2017*; *European Food Safety Authority, 2014*). However, *Salmonella* is not a normal flora in finfish and shellfish products. The major sources of *Salmonella* contamination in seafood include aquatic and aquaculture systems, seafood processing facilities, hygiene practices during transport, storage, and handling (*Amagliani, Brandi & Schiavano, 2012*; *Fernandes et al., 2018*). In this study, the prevalence of *Salmonella* ranged from 14% to 47%. This prevalence was similar to the prevalence of 25%

*Salmonella* contamination of shrimp cultured in Vietnam (*Phan et al., 2005*), but substantially less than the 90% to 100% prevalence of *Salmonella* contamination in fish (93%) and shrimp (100%) collected from a market in Indonesia (*Pramono et al., 2019*).

Type of seafood, sampling retail location, use of ice during retail display, regional source of seafood, and presence of *E. coli* and *Vibrio* were all significantly associated with the presence of *Salmonella* (Table 5). These risk factors should be interpreted with caution given the observational nature of this study and the possibility of undetected confounding in the statistical analysis. The presence of *E. coli* in a seafood sample was associated with a 4-fold increase in the odds of *Salmonella* contamination (OR = 4.0, $P < 0.0001$); similarly, the presence of *V. alginolyticus* in a seafood sample was associated with a 1.4-fold increase in the odds of *Salmonella* contamination (OR = 1.4, $P < 0.04$).

Seafood displayed on ice during retail had almost twice the odds of *Salmonella* contamination (OR = 1.7, $P < 0.0001$) than seafood not displayed on ice. This may seem counterintuitive, but ice used to chill seafood can be contaminated with pathogenic microorganisms (*Falcão, Falcão & Gomes, 2004*). Ice can be a vehicle for various pathogenic organisms, including diarrheagenic *E. coli*, *Aeromonas*, *S. enteritidis* and fecal coliforms (*Falcão et al., 2002*; *Falcão, Falcão & Gomes, 2004*; *Kirov, 1993*). In this study, most of the ice used to store seafood was at risk of rapidly melting due to high ambient temperatures in open air conditions. Melting ice can spread bacteria from one seafood item to nearby retail items, readily contaminating other seafood left standing in contaminated melt water.

In addition, the physical placement of seafood for display in retail markets can spread bacterial contamination between seafood items if seafood handlers do not practice proper sanitation during handling (*i.e.*, bare hands touching multiple seafood items; not replacing latex or plastic gloves at high enough frequency during retail display placement of seafood items). Therefore, maintaining sanitary conditions during the production, storage, and use of ice to prevent microbial contamination should be closely observed. Implementation of programs for food safety and also for prevention and control of diarrheal diseases have reduced mortality and morbidity rates of diarrheal diseases and strengthened food safety in Thailand (*Food Control Division, Food and Drug Administration, Thailand, 2004*).

Seafood sourced from Eastern Thailand had a 3.5-higher odds of *Salmonella* contamination than seafood from other regions (OR = 3.5, $P < 0.0001$). The coastal area of Eastern Thailand has concentrated areas of industrialization, agricultural development, and tourism-related urbanization, with major concerns of increased water pollution and resource depletion (*Nitivattananon & Srinonil, 2019*). Wastewater quality is a major concern for this area, especially in Chonburi and Rayong Provinces due to several industrial estates. Moreover, Chonburi, Chachoengsao, and Rayong Provinces have been designated for developing the Eastern Economic Corridor (EEC), so reduction of waste and wastewater is of increasing concern.

In this study, the diversity of *Salmonella* serovars varied between the different seafood products. Pacific white shrimp had the greatest diversity with 21 different serovars with prevalence per serovar ranging from 1–3%. Eleven serovars from isolated from oysters,

with prevalence per serovar ranging from 1–5% similar to Pacific white shrimp. In contrast, only 5 serovars were isolated from blood cockles, with a similar range of prevalence per serovar of 2–5%. Least diverse were isolates from Asian seabass. Only two serovars were recovered, with 52 of 56 of these *Salmonella* isolates being Matopeni and the remainder being Paratyphi B. Serotype Matopeni has been reported in aquatic pet shops (*Gaulin, Vincent & Ismail, 2005*) and in food supplements from Germany (*European Commission, 2018*). The infection of *S.* Matopeni has been reported in Malaysian children (*Lee et al., 2003*). *Salmonella* Paratyphi B in raw tuna sushi imported from Indonesia in 2015 caused 65 foodborne cases in the U.S. (*Centers for Disease Control & Prevention, 2018*). *S.* Typhimurium, *S.* Enteritidis, *S.* Typhi, and *S.* Paratyphi B were also detected in fresh fish in Iran (*Rahimi, Shakerian & Falavarjani, 2013*). *S.* Paratyphi B can be classified as d-tartrate fermenting (dT+) and d-tartrate non-fermenting (dT-) strains. The dT+ strain is less virulent and commonly reported with gastroenteritis, while the dT-strain is associated with paratyphoid fever. The dT+ strain is associated with a significant emerging disease worldwide and of public health concern (*Denny et al., 2007*; *Hassan et al., 2018*). Hence, classification of *S.* Paratyphi B biotype should be further investigated.

The common serovars in Pacific white shrimp were Itami (11%, *n* = 5/47) and Leith (9%, *n* = 4/47). Itami was first documented as a new serovar from a traveler to Thailand suffering from gastroenteritis (*Sakazaki et al., 1981*). Itami has also been reported from infected humans in Taiwan (*Kuo et al., 2014*). In contrast to serovars isolated during this study, serovars *S.* Weltevreden, *S.* Tennessee, and *S.* Dessau were isolated from shrimp from the Mekong Delta, Vietnam (*Phan et al., 2005*). The most common *S. enterica* serovar isolated from oysters was Give (19%, *n* = 8/43 isolates), which is different from oysters in the U.S. where Newport was the most common serotype (*Brands et al., 2005*). A previous study in Western Thailand found that the most common serovar in cultured oysters (*C. lugubris* and *C. belcheri*) from Phang Nga Province was Paratyphi B (*Jeamsripong, Chuanchuen & Atwill, 2018*). This suggests that the distribution of *Salmonella* serovars within Thailand depends on geographical location and type of seafood. Serovar Give is an enteric serotype usually isolated from swine and ruminants, but rarely found in humans (*Higgins et al., 1997*). Runoff from livestock feedlots near oyster farms could explain this contamination. *S. enterica* Give has been frequently reported in European national laboratories (*Jansen et al., 2005*). The higher virulence of the Give serovar compared to other non-typhoidal *Salmonella* may explain the higher hospital rate associated with human Give infections (*Girardin et al., 2006*).

Even though typhoid and paratyphoid salmonellosis are endemic diseases in Thailand, typhoid fever rates declined, and paratyphoid stabilized from 2003 to 2014 in this nation (*Techasaensiri et al., 2018*). In Thailand, *S. enterica* Weltevreden is commonly reported in human, frozen seafood, frozen ducks, and polluted water (*Bangtrakulnonth et al., 2004*). *S.* Weltevreden, *S.* Stanley, *S.* Anatum, and *S.* Rissen are frequently reported in human from northern and central Thailand (*Prasertsee et al., 2019*; *Sirichote et al., 2010*). Therefore, surveillance and monitoring of oysters due to this ~20% prevalence of

*Salmonella* contamination, and fully cooking oysters prior to consumption are both needed to reduce the risk of food-borne *Salmonella* infection from Thai-cultured oysters.

In this study, the most common *Salmonella* serovar found in blood cockles was Rissen (33%, $n$ = 8/24 isolates), similar to a study in India (*Kumar, Surendran & Thampuran, 2009*), but it should be noted that none of the five different serovars isolated from cockles had a prevalence above 5%. Seafood such as cockles can acquire *Salmonella* from contaminated water or other environmental matrices during aquaculture, processing, shipping, and retail display. Good hygiene and basic sanitation together with proper seafood handling and storage should be performed throughout the food chain (farm to fork) to reduce the risk of seafood-related *Salmonella*.

According to BQSF for Thailand seafood for human consumption should have no detectable *V. parahaemolyticus* and *V. cholerae* in 25 g of sample; however, 50–60% of samples contained these bacteria. This high prevalence is consistent with previous work demonstrating that between 2003 and 2015 the prevalence of *V. parahaemolyticus* was 64% in oysters, followed by clams (53%), fish (51%), and shrimp (48%) (*Odeyemi, 2016*). *V. parahaemolyticus*, *V. cholerae*, and *V. vulnificus* are important seafood-borne pathogens that cause gastroenteritis in humans. *V. alginolyticus* can cause ear infection and intestinal disease in humans. In this study, the main source of *V. parahaemolyticus* was blood cockles (OR = 9.1, $P < 0.05$), while *V. cholerae* was commonly found in Asian seabass (OR = 4.0, $P > 0.05$). *V. parahaemolyticus* and *V. vulnificus* have been reported in bivalves in many countries such as Thailand, China, and Korea (*Changchai & Saunjit, 2014*; *Jiang et al., 2019*; *Ryu et al., 2019*). In this study, shrimp and oysters were predominantly contaminated with *V. vulnificus* and *V. alginolyticus*.

*Clostridium perfringens*, *Staphylococcus aureus*, *V. parahaemolyticus*, and *Salmonella* spp. are the leading causes of foodborne illnesses in Thailand (*Bureau of Epidemiology, 2019*). In Thailand, human salmonellosis caused 167 illnesses per 100,000 persons in 2019. Contaminated produce and water have been indicated as important sources of *Salmonella* infection (*Bureau of Epidemiology, 2019*). The trend of *V. cholerae* infection decreased from 2.51 to 0.02 cases per 100,000 persons during 2010–2019, and contaminated water and seafood, poor sanitation, and dense housing have been blamed as sources of contamination (*Bureau of Epidemiology, 2019*).

In this study, the determination of bacterial prevalence and abundance was made using culture-dependent methods. For *Salmonella* spp. and *Vibrio* spp. detection this approach has high accuracy and sensitivity compared with certain molecular techniques (*Almeida et al., 2013*; *Eriksson & Aspan, 2007*; *Hara-Kudo et al., 2001*; *Mainar-Jaime et al., 2013*; *Yeung & Thorsen, 2016*). However, these methods can fail to detect viable but nonculturable state (VBNC) strains. VBNC bacteria can preserve metabolic activity and generate virulent proteins (*Alleron et al., 2013*; *Morishige, Fujimori & Amano, 2015*). Hence, molecular techniques are recommended to determine bacterial contamination, but unfortunately such equipment, needed supplies and training required to implement these molecular techniques are not readily available to many food safety officers, even in the developed world. In the absence of molecular equipment, the use of an appropriate MPN assay is justified and provides an estimate of bacterial load, along with ease of

interpretation of the results. In this study, the determination of bacterial species was made using bacterial phenotyping methods. This culture-dependent method was justified given our goal of developing a highly sensitive assay using readily available reagents.

*E. coli* concentrations appeared predicted *Salmonella* contamination of seafood. Based on the Youden index (*Ruopp et al., 2008*), a cutoff value for *E. coli* was $1.3 \times 10^4$ MPN/g can be implemented for both monitoring seafood for *Salmonella* contamination and to establish threshold control measures at processing or during retail storage. This cutoff is much higher than the microbiological criteria set forth in the Commission Regulation (EC) No 2073/2005 (*European Commission, 2005*), and the BQSF, Thailand (*Bureau of Quality & Safety of Food, 2020*). This reflects high concentrations of *E. coli* in this sample collection. Given that the detection of *Salmonella* and *Vibrio* spp. is of similar expense and technical difficulty as quantifying *E. coli* concentrations in seafood matrices, it may be more expeditious and more accurate to focus seafood safety monitoring protocols on *Salmonella* and *Vibrio* spp. detection rather than rely on indicator bacteria like *E. coli* that invariably suffer from false-positive and false-negative signals.

## CONCLUSIONS

Finfish and shellfish products sold in Bangkok are contaminated with diverse *Salmonella* serovars and species of *Vibrio*. Although the concentration of *E. coli* predicted *Salmonella* contamination for these seafood samples, the high cutoff value ($1.3 \times 10^4$ MPN/g) for maximal test accuracy will likely prevent this method from being adopted as a food hygiene surveillance tool. We recommend measuring *Salmonella* and *Vibrio* directly to assess the risk of food poisoning. Bacterial contamination varied by seafood commodity, with substantial differences between Asian seabass, oysters, blood cockle, and Pacific white shrimp. This may reflect different aquaculturing, harvesting, processing, and retail display practices.

## ACKNOWLEDGEMENTS

The authors thank Chailai Chareamchainukul, Mullika Kuldee, Varangkana Thaotumpitak, and Saweeyah Toodbat for their technical assistance.

### Funding

The work was financially supported by a grant from Chulalongkorn University-Veterinary Science Fund (RG 06/2562), Thailand. The funders had no role in study design, data collection and analysis, decision to publish, or preparation of the manuscript.

### Grant Disclosures

The following grant information was disclosed by the authors:
Chulalongkorn University-Veterinary Science Fund: RG 06/2562.

### Competing Interests

The authors declare that they have no competing interests.

## Author Contributions

- Edward R. Atwill analyzed the data, authored or reviewed drafts of the paper, and approved the final draft.
- Saharuetai Jeamsripong conceived and designed the experiments, performed the experiments, analyzed the data, prepared figures and/or tables, authored or reviewed drafts of the paper, and approved the final draft.

## Data Availability

The data are available in the Supplemental File.

## Supplemental Information

Supplemental information for this article can be found online at http://dx.doi.org/10.7717/peerj.12694#supplemental-information.

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
