# Peer review of "Bacterial diversity and potential risk factors associated with Salmonella contamination of seafood products sold in retail markets in Bangkok, Thailand"

_PeerJ, doi:10.7717/peerj.12694_

## Round 0.1 · original submission · Minor Revisions

I have a question about the references that should be addressed before I can send your manuscript off for review. Line 516 shows Martinez-Urtaza et al. (2005) J Food Prot 68:1077; however, that journal shows authors for that paper as Phan et al., which is confusing to me.

Also, I suggest the authors use reasonable significant figures. For example, line 26 presents the following "V. parahaemolyticus (58.81%), V. cholerae (48.96%), V. alginolyticus (18.51%)..." I think those percentages should be presented as (59%, 49%...).

Regards,

Michael

---

## Round 0.2 · Major Revisions

The reviews were mixed, including one recommendation to reject. I am not sure if the reviewers' concerns can be addressed without more data. At the very least, a resubmission should justify the use of bacterial phenotyping methods instead of molecular techniques. Reviewers also mentioned several grammatical errors, which suggests the need for revisions for format and style. To provide examples, I include an annotated draft.

Regards,

Michael

Reviewer 1 ·

Basic reporting

The structure of the article is adequate, as well as the tables and figures presented. The language is clear, and literature references support the findings and discussion.

Experimental design

The research presents a regional focus on the contamination of some seafood. However, the high contamination rates and testing the association of E. coli and Salmonella results bring insights that can be verified in other products in locations beyond Bangkok. Therefore, although it is a verification of the contamination of a regional product, the authors overcome this limitation of the study through data analysis and discussion of the paper. Furthermore, the methodology used was adequate and well described, the research question was clear and there was a rigorous investigation.

Validity of the findings

Data were well explored and properly analyzed. My only question is whether it would not be possible to use a T test to compare bacteria and different types of samples (Figure 4). I understand that due to the difference in sample N of each type of seafood, this comparison may be compromised. However, I suggest that in the next papers with this approach you carry out the sampling standardization between foods, or if this was the previous objective of this study, describe in the text why some products have a different N than others.

Moreover, a clear gap in the present paper, as mentioned in the conclusion, was the failure to verify the causes of this high contamination. However, although this matter has not been determined I believe that this work brings important information and will contribute to a better understanding of the contamination of these seafood.

Furthermore, I strongly advise you to carry out a more in-depth study trying to determine the causes of this high contamination, in order to obtain a microbiologically safe product.

Additional comments

I suggest the authors review the use of bacterial names in italics in the text. In conclusion section all bacterial names are not italicized.

·

Basic reporting

No main comment, limitations of statistical analysis could be discuss in the section Discussion and the results were not analyzed according to the period of sampling on several months.

Experimental design

Table 1 needs to be revised (Regional source of seafood)

Validity of the findings

Some confounding variables or interpretation bias could occur during the statistical analysis

Additional comments

See the additional review file

Reviewer 3 ·

Basic reporting

The English language should be improved some examples include lines 79, 96.

Authors should improve the information of the epidemiology of these pathogens in Thailand. They provide an extensive information about the seafood consumption and production.

Figure 1 should be improved, standard errors should be showed. VC, VP should be indicated. No figure captions were provided.

Experimental design

Authors should indicate if the analysis were performed in duplicate or triplicate.

L95- 99 Authors mention that individual seafood sample was purchased in the early morning (5 to 7 a.m.) and samples were kept on ice during transportation and submitted to the laboratory within 3 h. You should specify what temperature each sample was at the point of collection and how the samples were transported to the laboratory.

L149 Confirmation of Vibrios spp. should be determined by PCR to specify serogroups and pathogenicity factors. Furtheremore, Viable but nonculturable (VBNC) state strains in fish and seafood samples should be determined because this state is a survival strategy when bacteria suffer from stressed conditions such as low temperature. Bacteria in this state cannot be detected by culture-based methods, but they still maintain metabolic activity, and are even able to produce virulence proteins (Alleron et al., 2013; Morishige et al., 2015). It is a significant concern for food safety and public health.

L146 Specify the commercially antiserums (S&A Reagents Lab Ltd., Lat Phrao, Bangkok, Thailand) used to determine the serotype of Salmonella.

Validity of the findings

Authors should indicate that the average concentration of indicator bacteria occurrence is the general or total average concentration, as they are the averages of the population sampled.

The risk factors associated with Salmonella contamination, data shown in table 5, are not adequately explained. Authors should indicate, according to the OR and P values, if the variable represents a risk factor. Moreover, temperature and %RH were not considered as risk factors. However, authors mentioned that most of the ice used to store seafood was at risk of rapidly melting due to high ambient temperatures in open air conditions.

From my point of view, Conclusions are limited to the results, which were obtained by microbiological methodology without confirmation by molecular biology. Today, a large number of designed assay formats using these technologies are available commercially for the detection in foods of practically all major established pathogens and toxins, as well as of many emerging pathogens. Molecular biological assays are more specific, sensitive, and faster than conventional microbiological methods.

Additional comments

Authors should validate their findings and improve the interpretation of their results and discussions.

---

## Round 0.3 · Major Revisions

We asked two reviewers of v1 to review v2 but only one responded. To provide a timely decision, I acted as a second reviewer. From that perspective, I made a number of suggestions for style and format (see attached pdf). The extent of those changes suggests the need for major revisions. Other major issues include, lack of background justifying this study, particularly in Abstract and the discussion of methods.

In my opinion the text should be revised to explain the choice for using culturing to quantify bacteria and phenotyping to identify them. I share the concerns of an earlier reviewer who recommended rejection because these methods do not detect VBNC bacteria and are not as specific for strain identification. In v2, these issues (culturing and identification) are conflated. This is not appropriate. Many investigators use culture-dependent methods to count indicator bacteria and then use genotyping or other molecular techniques to identify strains. In the attached pdf, I suggest revisions how the methods reported in the submitted manuscript could be justified.

Reviewer 1 ·

Basic reporting

The authors corrected the paper as requested.

Experimental design

no comment

Validity of the findings

no comment

Additional comments

The authors satisfactorily answered the questions addressed.

---

## Round 0.4 · accepted · Accept

Reviewers to earlier versions of this manuscript declined the opportunity to review this version, but I think the manuscript is acceptable for publication. I did make several edits for form and style (see attached pdf) that can be addressed in production. Importantly, after reading the manuscript again it appears that you are recommending that Salmonella and Vibrio should be measured directly, rather than the fecal indicator bacteria. I changed your conclusions to state that.